# Factors Contributing to Traffic Accidents in Hospitalized Patients in Terms of Severity and Functionality

**DOI:** 10.3390/ijerph20010853

**Published:** 2023-01-02

**Authors:** Alexandra Carolina Canonica, Angelica Castilho Alonso, Vanderlei Carneiro da Silva, Henrique Silva Bombana, Aurélio Alberto Muzaurieta, Vilma Leyton, Júlia Maria D’Andrea Greve

**Affiliations:** 1Laboratory of Movement, Institute of Orthopedics and Traumatology, Clinics Hospital, Medicine School, University of Sao Paulo, Sao Paulo 04503010, Brazil; 2Graduate Program in Aging Sciences, Universidade São Judas Tadeu, Sao Paulo 03166000, Brazil; 3Department of Legal Medicine, Bioethics, Occupational Medicine and Physical Medicine and Rehabilitation, Medicine School, University of Sao Paulo, Sao Paulo 01246903, Brazil; 4Medical School, University of Michigan, Ann Arbor, MI 48109, USA

**Keywords:** external causes, traffic accidents, severity indices, disability, psychoactive substances

## Abstract

Trauma-related injuries in traffic-accident victims can be quite serious. Evaluating the factors contributing to traffic accidents is critical for the effective design of programs aimed at reducing traffic accidents. Therefore, this study identified which factors related to traffic accidents are associated with injury severity in hospitalized victims. Factors related to traffic accidents, injury severity, disability and data collected from blood toxicology were evaluated, along with associated severity and disability indices with data collected from toxicology on victims of traffic accidents at the largest tertiary hospital in Latin America. One hundred and twenty-eight victims of traffic accidents were included, of whom the majority were young adult men, motorcyclists, and pedestrians. The most frequent injuries were traumatic brain injury and lower-limb fractures. Alcohol use, hit-and-run victims, and longer hospital stays were shown to lead to greater injury severity. Women, elderly individuals, and pedestrians tend to suffer greater disability post-injury. Therefore, traffic accidents occur more frequently among young male adults, motorcyclists, and those who are hit by a vehicle, with trauma to the head and lower limbs being the most common injury. Injury severity is greater in pedestrians, elderly individuals and inebriated individuals. Disability was higher in older individuals, in women, and in pedestrians.

## 1. Introduction

The World Health Organization (WHO), according to the 2010 global road traffic accident report, instituted the “Decade of Action for Road Safety 2011–2020”, with the objective of reducing road deaths by 50%, a program to which Brazil was a signatory. Unfortunately, this goal was not achieved, so the “Decade of Action for Traffic Safety 2021–2030” was launched with the same objective [1]. The number of victims remains remarkably high with traffic and transportation-related accidents, creating a serious public health issue [2]. 

The 2018 WHO report showed an increase in the absolute number of deaths from traffic accidents worldwide, placing them among the ten leading causes of death globally [3]. Low-income countries suffer the highest number of traffic accidents compared to more economically advantaged countries [4].

In 2014, more than 50% of all deaths from traffic accidents among all countries in the Americas occurred in Brazil [4]. Between 2017 and 2022, São Paulo showed an upward trend in the death toll due to traffic accidents, but data on traffic accident morbidity are scarce [5].

Currently, literature on factors leading to and morbidity from traffic accidents is lacking. Most research uses data on mortality and the number of hospitalizations but does not elaborate on the relationship between causal factors and victims’ injury severity nor do they explore the resulting disabilities.

Recognizing and understanding the root causes, extent, and consequences of traffic accidents is the first steps toward developing effective and efficient prevention programs and policies.

The present study aimed to evaluate and identify the factors associated with hospitalizations due to traffic accidents at Hospital das Clínicas da Faculdade de Medicina da Universidade de São Paulo (HCFMUSP), as well as the relationship of these factors with injury severity at the time of occurrence as well as thirty days after discharge.

## 2. Materials and Methods

This is an epidemiological, descriptive, cross-sectional, prospective, and observational study approved on 18 May 2017, by the Ethics Committee of HCFMUSP under number 1240 that was realized in the Emergency Department of the *Instituto Central* of HCFMUSP. Written informed consent was obtained from the patients to publish this paper. The investigations were carried out following the rules of the Declaration of Helsinki of 1975, revised in 2013.

### 2.1. Subjects

The study included all traffic-accident victims treated at emergency departments and hospitalized at HCFMUSP between July and October 2017 (for the pilot study) and between August 2018 and July 2019. The inclusion criteria were individuals over 18 years old; they were victims of traffic accidents, as defined by the ICD-10, who received medical attention six or fewer hours after the accident, who required hospitalization and signed the Free and Informed Consent Form. Unconscious patients were included, and their family member(s) signed the Free and Informed Consent Form. Patients transferred to other hospitals in São Paulo, victims who died in emergency care and individuals who did not want to maintain their participation after the initial collection were excluded from the study.

All cases that met the inclusion criteria during the collection period were collected. The sampling method was one of convenience.

### 2.2. Procedures

#### 2.2.1. Collection of Patient Data


**Emergency department**


Patient identification and inclusion criteria

Collection of the RTS (Revisited Trauma Score)—a functional analysis of trauma severity—as performed. A lower score indicates a greater likelihood of death.

Blood collection for toxicology analysis—the collection team consisted of four nurses from the unit. The nurse that happened to be on duty identified the hospitalized victim and collected the blood, using an appropriate aseptic technique without the use of alcohol to clean the skin. The victim was identified by name, registration number, and inpatient unit. The sample was stored in a common refrigerator and then sent to the Toxicology Laboratory of the Department of Legal Medicine, Medical Ethics and Social and Work Medicine of FMUSP.


**Inpatient unit**


Interview—data were collected from all patients hospitalized at HCFMUSP due to traffic accident-related injuries who signed the Free and Informed Consent Form and were transferred from emergency departments. All interviews were conducted by the collection team on the inpatient unit. When direct contact with the patient was not feasible, family members were interviewed, and the patient was contacted later, either during hospitalization or after discharge.

Collection of severity indices was undertaken with the Abbreviated Injury Scale (AIS)/ Injury Severity Score (ISS). The ISS categorization was mild (1 to 9), moderate (9 to 15), severe (16 to 25) and very severe (greater than 25) [6].


**30 days after hospital discharge**


Functional independence measure (FIM) application:

This was conducted at home (discharged patients) by telephone interview or during an ambulatory patient follow-up.

A score of 104 or greater was used as a cutoff given that it defines the presence of complete functional independence. Values below 104 indicate some degree of dependence on the part of the patient [7].

#### 2.2.2. Toxicologic Analysis

Alcohol: blood alcohol level checks were performed by gas chromatography with a head-space separation technique using a methodology previously validated at the Toxicology Laboratory of the Department of Legal Medicine, Medical Ethics and Social and Work Medicine at FMUSP.

Drugs: psychoactive substances (illicit drugs) were quantified in blood samples at the Division of Forensic Sciences of the Norwegian Institute of Public Health, a European reference center for toxicologic analysis related to traffic. The methodology used involves liquid chromatography coupled to mass spectrometry, tandem system (LC-MS/MS). The following substances were screened for detection: MDMA (ecstasy), cocaine and marijuana.

For positive cases, the drug was confirmed.

#### 2.2.3. Collection

The collection was performed by venipuncture or other venous access available in the included patients. No type of alcoholic solution was applied to disinfect the collection site. Five milliliters of blood was collected in two vacutainer tubes containing fluoride/EDTA as an anticoagulant. One of the tubes was used for alcohol analysis, and the other was used for research/quantification of illicit drugs.

#### 2.2.4. Storage

The samples were sent to the Toxicology Laboratory of the Department of Legal Medicine, Medical Ethics and Social and Work Medicine at FMUSP and were frozen prior to performing the blood alcohol analysis.

For analysis of the other substances, the samples were sent in frozen batches to the Forensic Science Division of the Norwegian Institute of Public Health.

### 2.3. Statistical Analysis

Data were stored and analyzed using SPSS Statistics 22.0 for Windows (SPSS, Inc.).

Descriptive analysis of categorical variables is presented as frequencies and proportions. The Shapiro–Wilk test was used to verify whether the variables had a normal distribution. Comparisons of the mean values of the continuous variables, separated by cluster were performed using the Mann–Whitney U test. Associations between categorical variables were analyzed using Pearson’s Chi-square or Fisher’s Exact Test. Cluster and decision tree analyzes were performed in R software, version 4.2.1.

A significance level of 5% was used throughout the statistical analysis.

#### 2.3.1. Cluster Analysis

Clustering is an unsupervised learning method that assists professionals in discovering hidden patterns in a dataset [8]. The software packages cluster and extra fact were used. The K-means clustering algorithm was used to divide participants into groups based on their characteristics. K-means clustering is one of the most widely used clustering algorithms. This method partitions the data into clusters or groups so that data that have the same characteristics are grouped into the same cluster, and data that have different characteristics are grouped into other groups [9]. The algorithm requires the availability of two key operations on the data: first, a distance metric to compare a pair of data objects, and second, a way to compute a representative (centroid) for a given set of data objects. The K-means algorithm is made up of two different phases. In the first phase, the K centers (centroids) are selected randomly, while the second phase allocates each data point to the closest center. Thus, the data were converted into a Z-score and entered into the algorithm. Two clusters were maintained considering the homogeneity in the derived groups and the balance between the classes [10]. The group was examined to confirm the final number of clusters and whether the group was sufficiently large with adequate statistical power defined as at least 10% of the total sample. To determine the appropriate number of K values in K means, we used the silhouette score value of different ranges of clusters to determine the appropriate number of K values for the dataset. Furthermore, the cluster distance measurements were performed using Euclidean distances [11]. The analysis was performed in such a way that intracluster similarity was high and the intercluster similarity is low.

#### 2.3.2. Decision Tree

For the analysis and construction of the decision tree predictor model, the *rpart* (recursive partitioning and regression trees) algorithm was used and the outcomes defined for prediction were RTS (mortality), ISS (injury severity) and FIM (disability). To build the model, variables specifically related to mortality, injury severity, and disability in traffic accident victims were selected after considering clinical practice, the literature relevant to the topic, and research experience. The variables selected included age, sex, use of alcohol and drugs, days of hospitalization and schooling. The model with the decision tree algorithm identifies the factors with the greatest influence on the outcome under analysis based on the attributes that best divide the dataset [12]. The e1071 package was used to build the model with the following parameters defined: (1) cp = 0.019 (a complexity parameter that eliminates any division that does not improve the fit of the model), (2) maxdepth = 3 (minimum number of observations that must exist in a node), and (3) xval = 10 (number cross-validation).

## 3. Results

At the HCFM/USP Emergency Department, 1039 patients with injuries from external causes were treated from July to October 2017 and August 2018 to July 2019. A total of 609 (58.6%) patients were not included because they did not meet the inclusion criteria, or because they were discharged or transferred to another hospital; 15 (1.4%) patients refused to participate and 109 (9.9%) patients died. A total of 128 (12.3%) patients were considered victims of traffic accidents and thus met the inclusion criteria.

The sociodemographic and emergency care profiles of the 128 victims of traffic accidents are shown in Table 1. Medical attention prior to hospital arrival was rendered among 116 (90.6%) patients. Two (1.6%) victims reported not receiving this type of care and 10 (7.8%) individuals did not report receiving care prior to presenting at the hospital.

Figure 1 shows the type of accidents by the 128 victims of traffic accidents hospitalized at HCFMUSP.

The accident sites included expressways (n = 31/24.2%), low-speed roads (n = 25/19.5%), medium-speed roads (n = 17/13.3%), intersections (n = 11/8.6%), sidewalks (n = 7/5.5%), others (n = 9/7%), not informed (n = 28/21.9%).

Table 2 shows information on the use of safety equipment by the 128 victims of traffic accidents hospitalized at HCFMUSP.

Among the motorcyclists (n = 73), 20 (27.8%) were motorcycle couriers.

Ecstasy was not detected in the studied sample. The presence of alcohol (n = 34/26.6%), marijuana (n = 11/8.6%) and cocaine (n = 17/13.3%) were verified.

Figure 2 shows the two clusters identified by the K-means algorithm. Other subdivisions of the two groups did not result in better “intracluster” homogeneity. They also resulted in overlapping and very small clusters. Therefore, two clusters were ultimately maintained for grouping purposes. These two dimensions explain 64.8% of the variability of the analyzed data.

Table 3 shows the clinical characteristics of the victims by cluster. From the original dataset (n = 128), the following two large clusters were derived: Cluster 1 with 80 (62.5%) participants and Cluster 2 with 48 (37.5%) participants.

Table 4 shows the characteristics of care and the victims’ accidents by cluster.

Table 5 shows the profile of the victims’ alcohol and drug use by cluster.

Among the five variables selected for the construction of the decision tree (age, sex, alcohol and drug use, days of hospitalization and schooling), two are presented in the RTS prediction graph (age and length of hospitalization) with R^2^ = 0.24 (Figure 3), one is presented in the ISS prediction graph (length of hospitalization) with R^2^ = 0.08 (Figure 4), and three are presented in the FIM prediction graph (age, days of hospitalization and sex) with R^2^ = 0.21 (Figure 4).

## 4. Discussion

The traffic-accident victims included in this study were predominantly young, white men. 

The epidemiological profile of morbidity in the present study is similar to that of the rest of Brazil and the world based on the available data [13,14,15,16,17]. A 2017 study in the Brazilian state of Paraná showed that 76.5% of traffic accident victims were men aged 32.2 years, similar to other data from Brazil, Africa, Asia and Europe [4,5,13,14,18]. Factors such as behavior (drug use, speed) and exposure (greater use of motorcycles for transportation and work) explain these numbers and point to a need for more specific accident prevention programs [11,12,13]. Speed, a key causal factor in traffic-related accidents as identified by the WHO, is corroborated as such in the present study by the higher incidence of accidents on expressways [3]. 

It is also important to note that most victims were motorcyclists and pedestrians. This study evaluated only hospitalized victims with more serious injuries and showed the highest risk of accidents to motorcyclists and pedestrians. The predominance of accidents involving motorcyclists, however, is not consistent with international data. Studies from low- and middle-income countries show that motorcyclists, pedestrians, and cyclists are more vulnerable to traffic-related accidents [4,5,13,14,15]. However, studies from higher-income countries show that passengers or drivers of four-wheel vehicles suffer more frequently from traffic-related accidents [3]. These differences can be attributed to several factors: underreporting, socioeconomic differences, and the greater use of motorcycles in the poorest countries as a means of work and transport.

In Brazil, the use of motorcycles as a means of transportation has grown exponentially in the last 20 years, which, in tandem with the more recent economic and employment crises, has increased the number of motorcyclists who rely on this form of transport for work-related activities as a motorcycle courier or employee of an online delivery app [19].

Most motorcyclists in this study wore helmets regularly (98.6%), a finding likely attributable to laws requiring helmets and fines associated with violating these laws. However, the use of other types of personal protective equipment (jackets, boots, gloves) and protective equipment attached to the motorcycle (antenna and lower limb protector) was much less predominant. The uncommon use of boots (26.4%) and lower limb protectors (22.4%) shows the lack of understanding on the part of the motorcyclists of the importance of protecting the lower limbs, which are the most common site of injury in these accidents. 

Additionally, the lack of seat belt use (44.4%) in four-wheel vehicle occupants was observed despite being legally mandatory and finable. The lack of seatbelt usage is likely correlated with injury severity. These data reinforce the need for ongoing programs and campaigns of guidance and education on traffic safety. The WHO highlights that the use of safety items significantly reduces the severity of injuries and the number of deaths from traffic accidents [3].

Two main, distinct groups were identified in our study based on cluster analysis between which no intersection was definitively shown. Cluster 2 was observed as experiencing a more severe clinical condition, increased length of hospitalization, higher risk of death, greater injury severity, and more severe functional disability. In addition, Cluster 2 was shown to require more helicopter assistance than Cluster 1, an observation that indicates increased injury severity in this cohort. In Cluster 2, more patients whose mechanism of injury was being hit by a vehicle were observed, which may explain the worse clinical picture in this group.

There was no statistical significance of alcohol and drugs in the two groups in the cluster, despite alcohol having a similar value (*p* = 0.08/Table 5), demonstrating that Cluster 2 had more patients under the influence of alcohol than Cluster 1. This finding suggests that alcohol may play a role in the more severe clinical condition observed in Cluster 2. Interestingly, marijuana and cocaine did not show the same relationship, possibly because few patients were actually found to have used these drugs prior to their accident. 

Ponce et al. found that almost 50% of deaths from traffic accidents in São Paulo are a consequence of alcohol use [20]. Even low blood alcohol concentrations are capable of interfering with cognitive functions important for driving, namely attention and concentration, executive function, perception, psychomotor skills, reaction time, and vigilance. Alcohol impairs driving skills such as attending to traffic signs, detecting road hazards and braking the vehicle [21]. The use of alcohol is positively correlated with injury severity, which is consistent with data from the Brazilian and the international literature [22,23].

Most studies show that the risk of traffic-related accidents resulting in serious injury increases substantially in collisions involving individuals under the influence of illicit drugs [24,25]. Marijuana affects driving performance as it slows the driver’s reaction time to external stimuli [24]. Cocaine, by contrast, can lead to reckless driving in addition to overalertness and even nervousness [25]. This relationship was not observed in the present study, possibly due to their statistically low power in the setting of few patients having used such substances.

Traumatic brain injury (TBI) and lower-limb injuries (fractures) were the most frequently diagnosed conditions, findings consistent with other literature evaluating a similar population [14,15,16]. TBI is a serious injury that is quite common in traffic-related accidents, especially among motorcyclists and pedestrians. Lower-limb injuries are also very common because of the high frequency of motorcycle accidents [5,13,14]. 

Among the predictors of mortality, it was found that individuals with an increased length of hospitalization and over 43 years old had a higher risk of death than those under 43 years of age. Additionally, among the individuals with shorter lengths of hospitalization (less than 15 days), older individuals had a high risk of death than younger individuals. McCoy points out that the elderly suffer more deaths than younger people from traffic accidents [26].

Individuals aged 43 years or less and hospitalized between 15 and 17 days were observed to have a higher risk of death than those hospitalized for greater than 17 days. That younger patients hospitalized between 15 and 17 days had a higher risk of death could be attributable to the severity of injury to organic structures as related to the mechanism of trauma and the more generally reckless behavior observed among younger people, who more frequently suffer from traffic-related accidents [13,14,15].

Equally as important as the other aforementioned predictors of injury severity based on this study is the observation that an increased length of hospitalization was positively correlated with injury severity. Chalya et al. found that traffic accident victims with longer hospital stays had more severe injuries (TBI and limb fractures), which required a prolonged healing period. These data are similar to those of the present study [27].

In the FIM predictor model, older victims hospitalized between 12 and 38 days experienced greater functional disability than older victims with more than 38 days of hospitalization. This finding seems paradoxical and contradicts most of the current literature as the length of hospitalization is positively correlated with injury severity. However, it is possible that older patients recover less fully at post-hospital discharge than younger patients, which may explain this finding. It is worth noting that older people may require more care after discharge to achieve a fuller functional recovery. This seemingly paradoxical result might also be explained by the relatively low number of elderly patients observed with more than 38 days of hospitalization (5% of the sample, Figure 4). No studies were identified that specifically verified the relationship between length of hospitalization and degree of functional disability in older individuals. Rocha et al. analyzed traffic-accident victims over 20 years of age and reported that an increased length of hospitalization is positively correlated with the degree of functional disability and injury severity [28].

Among younger patients overall in this study, women had greater functional disability than men. However, data in the literature on the disability rate by sex in traffic accidents are controversial. Esiyok et al. found a higher frequency of disability among men than women; Ferrando found no difference between men and women; and Berecki-Gisolf found that female sex is a risk factor for disability, ostensibly due to women experiencing more pain [29,30,31].

Ultimately, the results of this study highlight the core behaviors that drive traffic-related accidents: alcohol, speed, lack of adherence to the use of recommended safety equipment, recklessness, and the overall riskier profile for motorcyclists and pedestrians. The ability to reduce morbidity and mortality due to traffic-related accidents depends on improving population-wide education on road safety as well as identifying and prosecuting drivers who pose a safety, and thus health, risk to the general population.

Public prevention policies need to be comprehensive, inclusive and permanent, as socioeconomic and educational inequality and access to employment and work forces a large proportion of young people to use motorcycles as their primary work tool, such as couriers or app delivery people.

A few limitations of the present study include cases lost to follow up in the emergency department, difficulty collecting information from patients suffering from the gravest injuries, and the omission or even falsification of information provided in the patients’ self-reports about their accidents.

## 5. Conclusions

The vast majority of traffic-accident victims hospitalized in a tertiary center are young, white men who were injured as pedestrians or motorcyclists. The most common injuries are traumatic brain injury and lower limb lesions.

Injury severity is greater in pedestrians, older people, those under the influence of alcohol, and those experiencing longer hospital stays. A shorter length of stay among young people is associated with a higher risk of death.

Disability is higher among older individuals with shorter hospital stays, women, and pedestrians.

## Figures and Tables

**Figure 1 ijerph-20-00853-f001:**
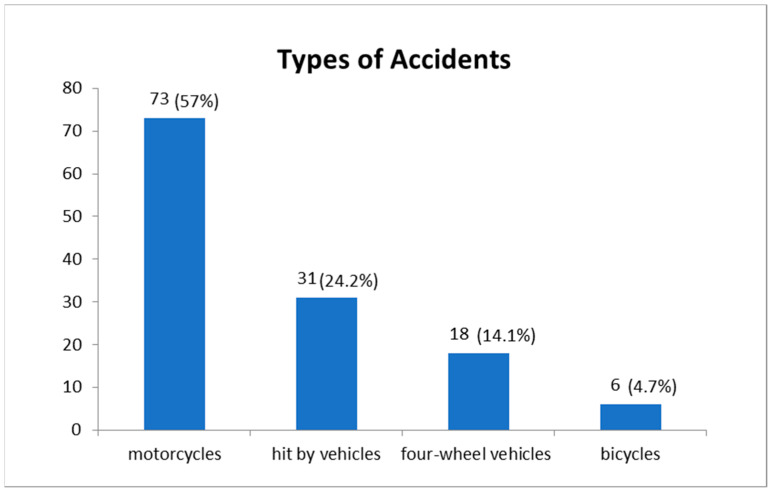
Type of accidents by the 128 victims of traffic accidents hospitalized at HCFMUSP.

**Figure 2 ijerph-20-00853-f002:**
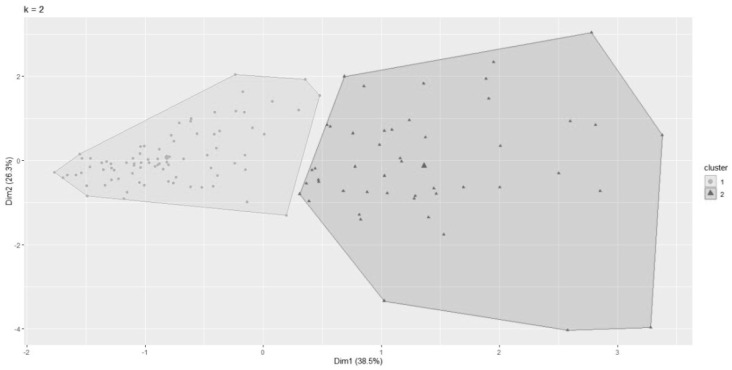
Cluster analysis of traffic accident victims hospitalized at HCFMUSP with the K-Means algorithm.

**Figure 3 ijerph-20-00853-f003:**
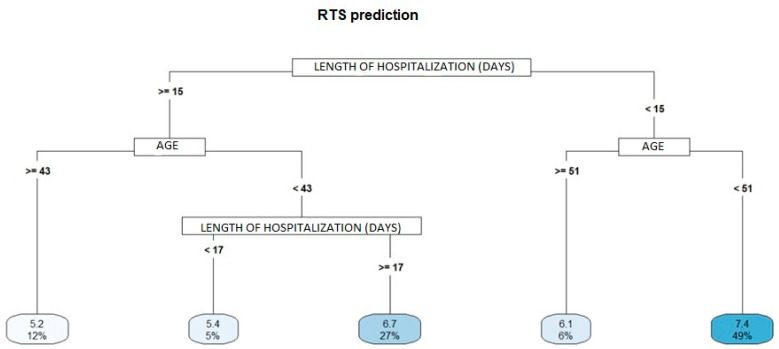
Decision tree to identify the predictive variables of the RTS (mortality).

**Figure 4 ijerph-20-00853-f004:**
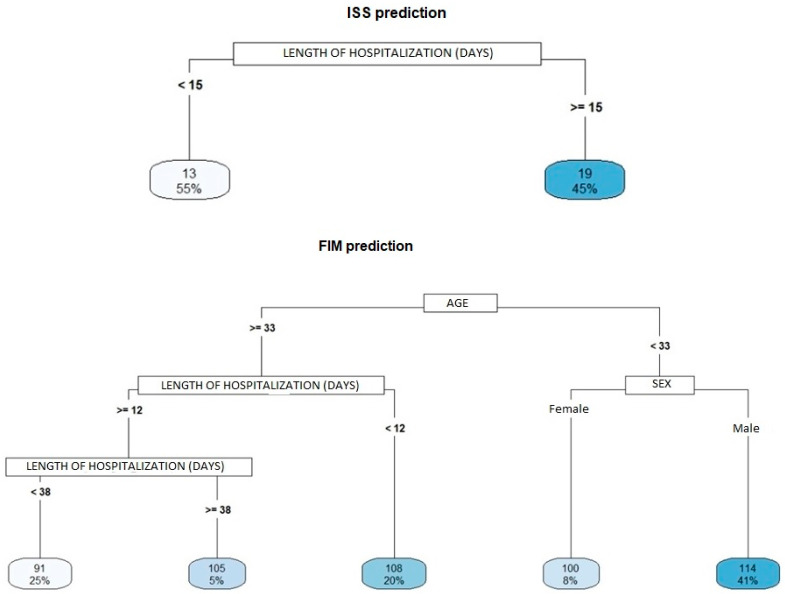
Decision tree to identify ISS predictive variables (injury severity) and FIM (disability).

**Table 1 ijerph-20-00853-t001:** Sociodemographic and emergency care profile of victims of traffic accidents treated at HCFMUSP.

**Age Group**	**Frequency**	**%**
18–29 years	53	41.4
30–39 years	40	31.3
40–49 years	18	14.1
50–59 years	11	8.6
>60 years	6	4.7
**Sex**	**Frequency**	**%**
Male	102	79.7
Female	26	20.3
**Skin Color**	**Frequency**	**%**
White	69	53.9
Mixed-Race	36	28.1
Black	18	14.1
Asian	3	2.3
Did not answer	2	1.6
**Type of Pre-hospital Care**	**Frequency**	**%**
SAMU	53	41.4
Firefighter	22	17.2
Águia	40	31.2
Other	2	1.6
Did not answer	11	8.6
**Diagnosis**	**Frequency**	**%**
Traumatic lower limb amputation	7	5.5
Traumatic upper limb amputation	1	0.8
Traumatic brain injury	34	26.6
Pelvic injury	3	2.3
Abrasions	38	29.7
Lower limb fracture	35	27.4
Fracture upper limbs	5	3.9
Other	5	3.9
**Secondary Diagnosis**	**Frequency**	**%**
Pelvic injury	2	1.6
Abrasions	15	11.7
Bruises	4	3.1
Lower limb fracture	30	23.4
Fracture upper limbs	14	10.9
Facial trauma	10	7.8
Thoracic trauma	5	3.9
Abdominal trauma	1	0.8
Other	4	3.1

SAMU: Ambulance, Águia: Helicopter.

**Table 2 ijerph-20-00853-t002:** Use of safety equipment by victims of traffic accidents treated at HCFMUSP.

**Motorcycle**	**N = 73**									
	Helmet		Boot		Jacket		Gloves		Antenna	
	Frequency	%	Frequency	%	Frequency	%	Frequency	%	Frequency	%
Yes	71	98.6	19	26.4	44	61.1	14	19.4	24	33.3
No	1	1.4	53	73.6	28	38.9	58	80.6	45	62.5
Uninformed									3	4.2
**Motorcycle**	**N = 73**									
	Lower-limb protector									
	Frequency	%								
Yes	16	22.2								
No	52	72.2								
Uninformed	4	5.5								
**Four Wheel Vehicle**	**N = 18**									
	Seat belt									
	Frequency	%								
Yes	6	33.3								
No	8	44.4								
Uninformed	4	22.2								

**Table 3 ijerph-20-00853-t003:** Clinical characteristics of traffic accident victims hospitalized at HCFMUSP by cluster.

Features	Cluster 1	Cluster 2	*p*-Value
	n = 80	n = 48	
Age (years)	32.9 (11.4)	37.75 (15.1)	0.09
Hospitality days	12.5 (11.8)	32.6 (40.1)	0.001
RTS (M/sd)	7.7 (0.53)	5.2 (1.6)	0.001
ISS (M/sd)	10.1 (5.3)	24.3 (10.5)	0.001
FIM (M/sd)	108.6 (16.9)	98.2 (24.1)	0.01

RTS: Revisited Trauma Score, ISS: Injury Severity Score, FIM: Functional independence measure, M (mean), sd (standard deviation).

**Table 4 ijerph-20-00853-t004:** Characteristics of care and accidents of victims of traffic accidents hospitalized at HCFMUSP by cluster.

Care Characteristics	Cluster 1 Frequency (%)	Cluster 2 Frequency (%)	X^2^ (*p*-Value)
**Type of care**			
SAMU	41 (57.7)	14 (30.4)	
Firefighter	12 (16.9)	10 (21.7)	
Águia	18 (25.3)	22 (47.8)	
Other	0 (0.0)	0 (0.0)	
			X^2^ = 8.9 (*p* = 0.01)
**ISS**			
Light	27 (33.7)	1 (2.1)	
Moderate	40 (50.0)	8 (16.7)	
Serious	12 (15.0)	21 (43.7)	
Very serious	1 (1.2)	18 (37.5)	
			X^2^ = 58.8 (*p* = 0.001)
**MIF**			
Dependency (FIM < 104)	20 (25.0)	19 (44.2)	
Independence (FIM ≥ 104)	60 (75.0)	24 (55.8)	
			X^2^ = 4.7 (*p* = 0.03)
**Accident characteristics**			
**Vehicle type**			
Motorcycle	52 (65.0)	21 (43.7)	
Four wheel vehicle	7 (8.7)	11 (22.9)	
Cyclist	3 (3.7)	3 (6.2)	
Hit by vehicle	18 (22.5)	13 (27.1)	
			X^2^ = 7.3 (*p* = 0.05)
**Description of the place**			
Crossing	8 (11.1)	5 (10.8)	
Low speed road	15 (20.8)	14 (30.4)	
Average speed road	17 (23.6)	5 (10.8)	
Expressway	17 (23.6)	16 (34.7)	
Sidewalk	8 (11.1)	3 (6.5)	
Other	7 (9.7)	3 (6.5)	
			X^2^ = 5.7 (*p* = 0.33)

SAMU: Ambulance, Águia: Helicopter, ISS: Injury Severity Score, FIM: Functional independence measure.

**Table 5 ijerph-20-00853-t005:** Profile of alcohol and drug use of victims of traffic accidents hospitalized at HCFMUSP by cluster.

Features	Cluster 1 Frequency (%)	Cluster 2 Frequency (%)	X^2^ (*p*-Value)
**Alcohol**			
Yes	17 (21.2)	17 (35.4)	
No	63 (78.7)	31 (64.5)	
			X^2^ = 3.1 (*p* = 0.08)
**Marihuana**			
Yes	8 (10.0)	3 (6.2)	
No	72.0 (90.0)	45 (93.7)	
			X^2^ = 0.5 (*p* = 0.53)
**Cocaine**			
Yes	8 (10.0)	9 (18.7)	
No	72 (90.0)	39 (81.2)	
			X^2^ = 1.9 (*p* = 0.15)

## Data Availability

The data can be found in the Appendix A of this article.

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
