# Peer review of "Factors Contributing to Traffic Accidents in Hospitalized Patients in Terms of Severity and Functionality"

_ijerph, 2023, doi:10.3390/ijerph20010853_

Round 1

Reviewer 1 Report

This study certainly has some interesting profiles but I believe that some methodological aspects need to be clarified before we can consider its publication.

I have one major concern about the inclusion criteria and the patients’ consent. The authors state that all enrolled patients signed the consent form to participate in the study. Does this mean that all patients unable to give consent (including minors) were excluded from the study? And the patients who had injuries so severe that they were unconscious were not taken into consideration? I believe that these aspects should be clarified, and the authors should specify whether or not this study considers major road injuries, in which victims usually arrive in hospitals unconscious or with severe impairment of consciousness.

Moreover, it is not clear to me how the toxicological analysis were conducted. In particular, were only screening or confirmatory analysis performed? Why were only 3 classes of substances analysed? Even opiates, benzodiazepines, some categories of drugs have a disabling effect on driving. It is not clear to me why the screenings were made only for MDMA, cocaine and marijuana.

Author Response

Point 1: This study certainly has some interesting profiles but I believe that some methodological aspects need to be clarified before we can consider its publication.

I have one major concern about the inclusion criteria and the patients’ consent. The authors state that all enrolled patients signed the consent form to participate in the study. Does this mean that all patients unable to give consent (including minors) were excluded from the study? And the patients who had injuries so severe that they were unconscious were not taken into consideration? I believe that these aspects should be clarified, and the authors should specify whether or not this study considers major road injuries, in which victims usually arrive in hospitals unconscious or with severe impairment of consciousness.

Response 1: Underage patients were not included and unconscious patients were included and the family signed the Free and Informed Consent Form.

Point 2: Moreover, it is not clear to me how the toxicological analysis were conducted. In particular, were only screening or confirmatory analysis performed? Why were only 3 classes of substances analysed? Even opiates, benzodiazepines, some categories of drugs have a disabling effect on driving. It is not clear to me why the screenings were made only for MDMA, cocaine and marijuana.

Response 2: All toxicological analyzes were screening and in positive cases confirmation was made. More than 30 substances were analyzed, including alcohol, marijuana, buprenorfin, diazepam, oxazepam, clonazepam, alprazolam, nitrazepam, Flunitrazepam, Diclazepam, zopiclon, zolpidem, morphine, codeine, 6-monoacetylmorphine, oxycodone, methadone, lamotrigine, amitriptyline, pregabalin, gabapentin, levomepromazine, quetiapine, hydroxyzine, alimemazine, cocaine, benzoylecgonine, cocaethylene, anhydroecgonine methyl ester, amphetamine, methamphetamine, ecstasy, fentanyl, tramadol.

It was not reported in the article as most substances were not found in the blood samples. Observe in the tables with all the results the incidence of use of other drugs. The drugs chosen are the most used and therefore kept in the article.

Reviewer 2 Report

Check the title for extraneous dashes. 

Grammatical mistakes need correcting.  In the abstract alone, "assess" should be inserted between "to" and "traffic" in the first sentence; "to injuries seriousness" should be replaced by "with serious injuries" in the third sentence.  "It was evaluated factors related to traffic accidents" should be replaced by "Factors related to traffic accidents that were evaluated included" in the fourth sentence; "serious" should be inserted before "injuries" in the sixth sentence; and in the seventh sentence "accidents" should replace "victims" and "shown to lead to" should be replaced with "associated with."  

RTS (p. 2), ISS & FIMS (p.4) need to be spelled out when first appearing in the text and not just in the footnote to Table 3.  

The study design is good and the write-up is also good and well-referenced. 

the cluster analysis needs further explanation.  

I would suggest the inclusion of a bar chart for types of vehicles.  

The exclusion of dead victims and those who transferred to other hospitals represents a limitation of the study.  

Author Response

Point 1: Check the title for extraneous dashes. 

Response 1: Typing error that will be fixed

Point 2: Grammatical mistakes need correcting.  In the abstract alone, "assess" should be inserted between "to" and "traffic" in the first sentence; "to injuries seriousness" should be replaced by "with serious injuries" in the third sentence.  "It was evaluated factors related to traffic accidents" should be replaced by "Factors related to traffic accidents that were evaluated included" in the fourth sentence; "serious" should be inserted before "injuries" in the sixth sentence; and in the seventh sentence "accidents" should replace "victims" and "shown to lead to" should be replaced with "associated with."  

Response 2: Corrections have been made, please check new text with corrections.

Point 3: RTS (p. 2), ISS & FIMS (p.4) need to be spelled out when first appearing in the text and not just in the footnote to Table 3.

Response 3: Corrections have been made. 

Point 4: the cluster analysis needs further explanation.

Response 4: The use of cluster analysis was made for the formation of more homogeneous groups, since the regular statistical analysis did not clearly point out the actors most related to traffic accidents. The formation of groups and the ranking of data helps to clarify the results, allowing a better assessment of the factors that may be related to traffic accidents. Also the separation into groups allows a better analysis of all the variables and how they relate to each other.

Point 5: I would suggest the inclusion of a bar chart for types of vehicles.

Response 5: the suggestion has been accepted and the graphic was included

Point 6: The exclusion of dead victims and those who transferred to other hospitals represents a limitation of the study.  

Response 6: the transfer to other locations could be a limitation, due to the reduction of the sample size, but it is important to emphasize that the transferred cases are less serious than the ones that remained. Thus, the current study evaluates serious victims hospitalized in a highly complex trauma center, that is, a more specific study of severe cases. The excluded deaths were those that occurred at the time of care and could not be evaluated later in terms of disability.

Round 2

Reviewer 1 Report

I think that this version of the manuscript is improved and Authors have addressed all my comments.

Best regards